# V^2^ReID: Vision-Outlooker-Based Vehicle Re-Identification

**DOI:** 10.3390/s22228651

**Published:** 2022-11-09

**Authors:** Yan Qian, Johan Barthelemy, Umair Iqbal, Pascal Perez

**Affiliations:** 1SMART Infrastructure Facility, University of Wollongong, Wollongong 2500, Australia; 2NVIDIA, Santa Clara, CA 95051, USA

**Keywords:** vehicle re-identification, Vision Outlooker, explainable AI, secure AI, smart cities

## Abstract

With the increase of large camera networks around us, it is becoming more difficult to manually identify vehicles. Computer vision enables us to automate this task. More specifically, vehicle re-identification (ReID) aims to identify cars in a camera network with non-overlapping views. Images captured of vehicles can undergo intense variations of appearance due to illumination, pose, or viewpoint. Furthermore, due to small inter-class similarities and large intra-class differences, feature learning is often enhanced with non-visual cues, such as the topology of camera networks and temporal information. These are, however, not always available or can be resource intensive for the model. Following the success of Transformer baselines in ReID, we propose for the first time an outlook-attention-based vehicle ReID framework using the Vision Outlooker as its backbone, which is able to encode finer-level features. We show that, without embedding any additional side information and using only the visual cues, we can achieve an 80.31% mAP and 97.13% R-1 on the VeRi-776 dataset. Besides documenting our research, this paper also aims to provide a comprehensive walkthrough of vehicle ReID. We aim to provide a starting point for individuals and organisations, as it is difficult to navigate through the myriad of complex research in this field.

## 1. Introduction

The goal of vehicle re-identification (ReID) is to retrieve a target vehicle across multiple cameras with non-overlapping views from a large gallery, preferably without the use of license plates. Vehicle ReID can play key roles in intelligent transportation systems [1], where the performance of dynamic traffic systems can be evaluated by estimating the circulation flow and the travel times, in urban computing [2], by calculating the information of origin–destination matrices, and in intelligent surveillance to quickly discover, locate, and track the target vehicles [3,4]. Some practical applications for vehicle ReID include: vehicle search, cross-camera vehicle tracking, automatic toll collection (as an alternative to expensive satellite-based tracking or electronic road pricing (ERP) systems), parking lot access, traffic behaviour analysis, vehicle counting, speed restriction management systems, and travel time estimation, among others [5]. With the widespread use of intelligent video surveillance systems, the demand for vehicle ReID is growing.

However, ReID can be very challenging due to pose and viewpoint variations, occlusions, background clutter, or small inter-class similarities and large inter-class differences. Two vehicles from different manufacturers might look very similar, whereas the same vehicle can appear very different from various perspectives; see Figure 1.

There are two types of ReID: open-set ReID and closed-set ReID. First, let us imagine a vehicle, which we refer to as the query, driving around in the large population centre of Sydney, Australia. Any time it drives past the field of view of a camera, a picture is taken by that camera. In a closed world, the query vehicle is known to the network, meaning that images of that vehicle already exist in the database, called the gallery. The goal of the model is then to re-identify the query vehicle in the gallery. This is performed by yielding a ranking of vehicle IDs the model thinks are the most similar to our query vehicle. Now, let us imagine that we have a visiting driver from Wollongong driving to Sydney for his/her first time. That vehicle is going to be new to the network. The closed-set re-identification model is not capable of identifying that new query, as the car does not exist in the database yet. Hence, this model is very limited and cannot be used for real-life applications. Open-set ReID is able to tackle this problem by first verifying if the newly registered vehicle is actually a new vehicle (verification task). If it is a new vehicle, then the new ID is added to the gallery. Else, if it is an already-seen vehicle, the model re-identifies which vehicle ID it corresponds to from the gallery (re-identification task). Open-set ReID applies to real-life scenarios, but is more difficult to solve. Unlike person ReID, where few works tackle the issue of open-set ReID, no one has attempted to tackle open-set vehicle ReID yet. Without license plate recognition, how can we recognize if it is an already-seen or a never-seen vehicle? If *we* do not know how to achieve this specific task, how can we teach an *AI* to do so?

Open-set ReID includes a verification and re-identification step; hence, it is closely related to closed-set ReID. Though we do not have the right tools presently to produce an open-set vehicle ReID model, we can build a stable and accurate closed-set ReID model, such that the final step would be to only focus on the verification task to solve open-set vehicle ReID. Therefore, our paper focuses on closed-set ReID. Let $X={\left\{x_{i},y_{i}\right\}}_{i=1}^{N}$ be a set of *N* training samples, where $x_{i}$ is an image sample of a vehicle and $y_{i}$ is its identity label. The ReID algorithm learns a mapping function f:X→F, which projects the original data points in X to a new feature space F. In this new feature space, the intra-class distance should be shrinked, while the inter-class distance should be as large as possible. Let q be a query image and $G={\left\{g_{j}\right\}}_{j=1}^{m}$ the gallery set. The ReID algorithm computes the distance between f(q) and every image in *G* and returns the images with the smallest distance. The gallery image set and the training image set should not overlap, i.e., the query vehicle should not appear in the training set. Vehicle ReID can thus also be regarded as a zero-shot  problem, distinguishing it from general image retrieval tasks [6].

When attempting to re-identify a vehicle, we would first focus on global information (descriptors), such as the colour or model. However, because of appearance changing under various perspective, global features lose information on crucial details and can, therefore, be unstable. Local features, on the other hand, provide stable discriminative cues. Local-features-based models divide an image into some fixed patches. Wang et al. [7] generated orientation-invariant features based on 20 different vehicle key point locations (front right tyre, front left tyre, front left light, etc.). Liu et al. [8] extracted local features based on three evenly separated regions of a vehicle. He et al. [9] detected window, lights, and the brand for each vehicle through a YOLO detector. Local descriptors include logos, stickers, or decorations. See the bottom row of Figure 1, where globally speaking, the three images look like the same red car, but locally, there are small, but crucial differences in the windshield.

Besides only visual cues, ReID models can also include underlying spatial or temporal aspects of the scene. Vehicle ReID methods can be classified into contextual and non-contextual methods. Non-contextual approaches rely on the appearances of the vehicles and measure the visual similarities in order to establish correspondences. Contextual approaches use additional information in order to improve the accuracy of the system. Commonly used side information includes: different views of the same vehicle; camera-related information, such as the parameters, location, or topology; temporal information such as the time between observations; license plate number; modelling changes in illumination, vehicle speed, or direction. Contextual information is used in the association stage of the vehicles. It is used to reduce the search space. Our method is a non-contextual method. We are interested in seeing how far we can push a ReID model without adding any side information.

Much research has been conducted on vehicle ReID, yet the number of papers is much lower when compared to their person ReID counterparts. Moreover, existing papers either are often very complex to understand or the code is not provided, making it harder for those wanting to get started. Our aim is, foremost, to document our research in a accessible way and to show that it is possible to create a successful model achieving 80.31% mAP on the VeRi-776 dataset by only using visual cues, resulting in the best scores to the best of our knowledge. Our code is available at: https://github.com/qyanni/v2reid (accessed on 14 September 2022).

In summary, the main contributions of this work are the following:We applied for the first time the VOLO [10] architecture in vehicle re-identification and show that attending neighbouring pixels can enhance the performance of ReID.We evaluated our method on a large-scale ReID benchmark dataset and obtained state-of-the-art results using only visual cues.We provide an understandable and thorough guide on how to train your model by comparing different experiments using various hyperparameters.

Section 2 introduces vehicle ReID, as well as the existing methods and research using convolutional neural networks. We present V^2^ReID in Section 3, as well as existing research based on Transformer-based methods. After going through the datasets and evaluation metrics in Section 4, we lay out the different experiments by fine-tuning several hyperparameters in Section 5, before concluding.

## 2. Related Work

Various methods exist for building a vehicle ReID model. We briefly go through these methods, with a deeper focus on attention-based methods, a key ingredient to our model.

### 2.1. A Brief History of Re-Identification

Compared to over 30 review papers on person re-identification, only four reviews on vehicle re-identification have so far been published (Table 1). Existing reviews broadly split the methods into sensor-based and vision-based methods. ReID methods based on sensors, e.g., magnetic sensors, inductive loops, GPS, etc., are not covered in this paper. Please refer to the surveys for explanations. Vision-based methods can be further broken down into hand-crafted-feature-based methods (referred to as traditional machine-learning-based methods [11]) and deep-feature-based methods. Hand-crafted features refer to properties that can be derived using methods that consider the information that is present in the image itself, e.g., edges, corners, contrast, etc. However, these methods are very limited to the various colours and shapes of vehicles. Other important information, such as special decorations or license plates, are difficult to detect because of the camera view, low resolution, or poor illumination of the images. This leads to very poor generalization abilities. Due to the success of deep learning in computer vision [12], convolutional neural networks (CNNs) were then introduced in the re-identification task to extract deep features (features extracted from the deep layers of a CNN).

Roughly speaking, deep-feature-based methods can be split into two parts: feature representation and metric learning (ML) [15]. Feature representation focuses on constructing different networks to extract features from images, while metric learning focuses on designing different loss functions. Feature representation methods can be further split into local  features (LFs), representation  learning (RL), unsupervised  learning (UL), and attention- mechanism (AM)-based methods (Figure 2). Their description, as well as their advantages and disadvantages are detailed in Table 2. Table 3 summarizes some sample works using these methods and their performances.

### 2.2. Attention Mechanism in Re-Identification

“In its most generic form, attention could be described as merely an overall level of alertness or ability to engage with surroundings.”[16]

Attention can be formed by teaching neural networks to learn what areas to focus on. This is performed by identifying key features in the image data using another layer of weights. When humans try to identify different vehicles, we go from obvious to subtle. First, we determine coarse-grained features, e.g., car type, and then identify the subtle and fine-grained level visual cues, e.g., windshield stickers.

Two types of attention exist: soft attention, e.g., SCAN [17], and hard attention, e.g., AAVER [18]. Generally speaking, soft attention pays attention to areas or channels and is differentiable. The latter refers to all the attention terms and the loss function being a differentiable function with respect to the whole input. Hence, all the weights of the attention can be learned by calculating the gradient during the optimization step [19]. Hard attention focuses on points [20], that is every point in the image is likely to extend the attention.

Table 3 presents a few works using the attention mechanism and their performance on the VeRi or the Vehicle-ID datasets. Other mentionable methods from the AI City Challenge [21] include SJTU (66.50% mAP) [22], Cybercore (61.34% mAP) [23], and UAM (49.00% mAP) [24].

**Table 3 sensors-22-08651-t003:** Summary of some results on vehicle re-identification in a closed-set environment using CNNs on the VeRi-776 [3] and Vehicle-ID [25] datasets.

Method	Year	Model	VeRi-776		Vehicle-ID (mAP (%)/R-1 (%))		
			mAP (%)	Rank-1 (%)	S	M	L
LF	2017	OIFE [7]	48.00	89.43	-	-	67.00/82.90
	2018	RAM [8]	61.50	88.60	75.20/91.50	72.30/87.00	67.70/84.50
	2019	PRN + RR [9]	74.30	94.30	78.40/92.30	75.00/88.30	74.20/86.40
ML	2017	Siamese-CNN + PathLSTM [26]	58.27	83.49	-	-	-
	2017	PROVID [27]	53.42	81.56	-	-	-
	2017	NuFACT [27]	48.47	76.76	48.90/69.51	43.64/65.34	38.63/60.72
	2018	JFSDL [28]	53.53	82.90	54.80/85.29	48.29/78.79	41.29/70.63
	2019	VANet [29]	66.34	89.78	88.12/97.29	83.17/95.14	80.35/92.97
	2020	MidTriNet + UT [30]	-	89.15	91.70/97.70	90.10/96.40	86.10/94.80
AM	2018	RNN-HA [31]	56.80	74.79	-	-	-
	2018	RNN-HA (ResNet + 672) [31]	-	-	83.8/88.1	81.9/87.0	81.1/87.4
	2019	AAVER [18]	61.18	88.97	74.69/93.82	68.62/89.95	63.54/85.64
	2020	SPAN w/ CPDM [32]	68.90	94.00	-	-	-
UL	2017	XVGAN [33]	24.65	60.20	52.89/80.84	-	-
	2018	GAN + LSRO + re-ranking [34]	64.78	88.62	86.50/87.38	83.44/86.88	81.25/84.63
	2019	SSL + re-ranking [35]	69.90	89.69	88.67/91.92	88.13/91.81	86.67/90.83

## 3. Proposed V${}^{2}$ReID

In the following section, we dive into the Transformer architecture for computer vision. Note that Transformer is a feedforward-neural-network-based architecture with an encoder–decoder structure, which makes use of an attention mechanism, in particular a self-attention operation. In other words, Transformer is the model, while the attention mechanism is a technique used by the model.

We used VOLO [10], a Transformer-based architecture, as the backbone of our model named V${}^{2}$ReID. We detail Transformer and VOLO as much as possible in the next paragraphs, as well as introduce the loss functions, evaluation methods, and dataset used in our process.

### 3.1. Rise of the Transformers

The development of deep-feature-based methods has gone through different stages. Early methods applied pure CNNs as their backbones to learn features, such as VGGNet [36] (*DLDR* [25]), GoogleLeNet [37] (*NuFACT* [27]), AlexNet [12] (*FACT* [3]), or ResNet [38] (*RNN-HA* [31]).

One shortcoming of convolution is that it operates on a fixed-sized window, meaning it is unable to capture long-range dependencies. Methods using self-attention can alleviate this problem as, instead of sliding a set of fixed kernels over the input region, the query, key, and value matrices are used to compute the weights based on input values and their positions. With the rise of Transformers revolutionizing the field of NLP [39], models based on the Transformer architecture have also gained more and more attention in computer vision. Among other models, Vision Transformer (ViT) [40] and Data-Efficient image Transformer (DeiT) [41] have stood out by achieving state-of-the-art results. It is clear that they also attract interest in re-identification. Before diving into Transformer-based vehicle re-identification, let us first explain what Transformers are.

The original Transformer [39] is an attention-based architecture, inheriting an encoder–decoder structure. It discards entirely the recurrence and convolutions by using multi-head  attention  mechanisms (MHSAs) (Figure 3) and pointwise  feed-forward networks (FFNs) in the encoder blocks. The decoder blocks additionally insert cross-attention modules between the MHSA and FFN. Generally, the Transformer architecture can be used in three ways [42]:1.Encoder–decoder: This refers to the original Transformer structure and is typically used in neural machine translation (sequence-to-sequence modelling).2.Encoder-only: The outputs of the encoder are used as a representation of the input. This structure is usually used for classification or sequence labelling problems.3.Decoder-only: Here, the cross-attention module is removed. Typically, this structure is used for sequence generation, such as language modelling.

Inspired by the vanilla architecture, researchers in computer vision have employed Transformer-like architectures for *classification* (ViT [40], DeiT [41]), *detection* (DETR [43], YOLOS [44]), *segmentation* (SETR [45], SegFormer [46]) and *object re-identification* (TransReID [47]). Visual Transformers can be as effective as their CNN counterparts on feature extraction for image recognition. For more information on Transformers in computer vision, please refer to the surveys [42,48,49,50,51,52].

### 3.2. Transformer in Vision

In the case of computer vision, Transformer has an encoder structure only. The following paragraphs detail how an input image is reshaped to what lives in the Transformer block. We try to detail each step as much as possible. Please refer to Figure 4 for the explanations.

#### 3.2.1. Reshaping and Preparing the Input

The vanilla Transformer model [39] was trained for the machine translation task. While the vanilla Transformer accepts sequential inputs/words (1D token embeddings), the encoder in the vision Transformer takes 2D images, which are split into patches. These are treated the same way as tokens (words) in an NLP application.

Patch  embeddings: Let **x** $\in R^{H\times W\times C}$ be an input image, where (H,W) is the resolution of the original image and *C* is the number of channels. First, the input image is divided into non-overlapping patches [40], which are then flattened to obtain a sequence of vectors:$$P\left(x\right)=\left[x_{p}^{1},x_{p}^{2},...,x_{p}^{N}\right]\in R^{N\times (P^{2}C)},$$
where $x_{p}^{i}\in R^{\left(P^{2}\cdot C\right)}$ represents the *i*th flattened vector, *P* the patch size, and $N=\frac{HW}{P^{2}}$ the resulting number of patches. The output of this projection of the patches is referred to as *patch embedding*. Example: Let an input be of dimension (256, 256, 3) and a patch size of (16×16). That image is divided into N=256 patches, where each patch is of dimension (16, 16, 3).

Sometimes, we can lose local neighbouring structures around the patches when splitting them without overlap. TransReID [47] and PVTv2 [53] generate patches with overlapping pixels. Let the step size be *S*, then the area of where two adjacent patches overlap is of shape (P−S)×P. The number of resulting patches is in total $N=\left\lfloor \frac{H+S-P}{S}\right\rfloor \times \left\lfloor \frac{W+S-P}{S}\right\rfloor$. A comparative figure (Figure 5) and the PyTorch-style commands (Algorithm 1) are provided.
**Algorithm 1: **
PyTorch-style command for non-overlapping vs. overlapping patches.
# non-overlapping patchesself.proj = nn.Conv2d(in_chans, embed_dim, kernel_size=patch_size, stride=patch_size)# overlapping patchesself.proj = nn.Conv2d(in_chans, embed_dim, kernel_size=patch_size, stride=stride_size)

Classification  [cls]  token: Similar to BERT [54], a learnable classification token [cls] is attached with the patch embeddings. This token aggregates the global representation of the sequence into a single vector. The latter then serves as the input for the classification task.

Positional  encoding: In order to retain the positional information of an entity in a sequence (ignored by the encoder, as there is no presence of recurrence or convolution), a unique representation is assigned to each token or patch to maintain their order. These representations are 1D learnable positional  encodings. The joint embeddings are then fed into the encoder.

#### 3.2.2. Self-Attention

In the Transformer Encoder block lives the multi-head attention, which is just a concatenation of single-head attention blocks.

Single-head  attention  block: Let an input **x** $\in R^{n\times d}$ be a sequence of *n* entities $\left(x_{1},x_{2},\ldots ,x_{n}\right)$ and *d* the embedding dimension to represent each entity. The goal of self-attention is to capture the interaction amongst all *n* entities by encoding each entity in terms of the global contextual information. Self-attention captures *long-term* dependencies between sequence elements as compared to conventional recurrent models. This is performed by defining three learnable linear weight matrices $W^{Q}\in R^{d\times d_{q}}$, $W^{K}\in R^{d\times d_{k}}$, and $W^{V}\in R^{d\times d_{v}}$, where $d_{q}=d_{k}$ and $d_{v}$ denote the dimensions of queries/keys and values.

The input sequence **x** is then projected onto these weight matrices to create query $Q={\mathrm{xW}}^{Q}$, key $K={\mathrm{xW}}^{K}$, and value $V={\mathrm{xW}}^{V}$. Subsequently, the output **Z**$\in R^{n\times d_{v}}$ of the self-attention layer is calculated as follows: (1)$$Z=\mathrm{Attention}\left(Q,K,V\right)=\underset{\mathrm{attention}\;\mathrm{matrix}}{\underset{︸}{\mathrm{SoftMax}\left(\frac{QK^{T}}{\sqrt{d_{k}}}\right)}}V.$$

The scores $QK^{T}$ are normalized with $\sqrt{d_{k}}$ to alleviate the gradient vanishing problem of the SoftMax function. In general terms, the attention function can be considered as a mapping between a query and a set of key–value pairs, to an output. The query, value, and key concepts are analogous to retrieval systems. for instance, when searching for a video (query), the search engine maps the query against a set of results in the database, based on the title, description, etc. (keys), and presents the best-matched videos (values).

Multi-head attention  block: If only a single-head self-attention is used, the feature sub-space is restricted, and the modelling capability is quite coarse. A multi-head self-attention (MHSA) mechanism linearly projects the input into multiple feature sub-spaces where several independent attention layers are used in parallel, to process them. The final output is a concatenation of the output of each head.

#### 3.2.3. Transformer Encoder Block

In general, a residual architecture is defined as a sequence of functions, where each layer *l* is updated in the form of
$$x_{l+1}=g_{l}\left(x_{l}\right)+R_{l}\left(x_{l}\right).$$

Typically, the function $g_{l}$ is the identity, and $R_{l}$ is the main building block of the network. With ResNet [38], residual architectures are more used in computer vision. They are easier to train and achieve better performance. The Transformer encoder consists of alternating layers of multi-headed self-attention (MHSA) and multilayer perceptron (MLP) blocks. LayerNorm (LN) is applied before every block [55], followed by residual connections after every block, in order to build a deeper model. Each Transformer encoder block/layer can then be written as
(2)$$\begin{array}{cc} Z_{l}^{{}^{\prime }} & =\mathrm{MHSA}\left(\mathrm{LN}\left(Z_{l}\right)\right)+Z_{l}\\ Z_{l+1} & =\mathrm{MLP}\left(\mathrm{LN}\left(Z_{l}^{{}^{\prime }}\right)\right)+Z_{l}^{{}^{\prime }} \end{array}$$
where MHSA(·) denotes the MHSA module and LN(·) the layer normalization operation. It is worth mentioning that this follows the definition of the vanilla Transformer, except that a residual connection is applied around each sub-layer, followed by LayerNorm, i.e.,
$$\begin{array}{cc} Z_{l}^{{}^{\prime }} & =\mathrm{LN}(\mathrm{MHSA}\left(Z_{l}\right)+Z_{l})\\ Z_{l+1} & =\mathrm{LN}(\mathrm{FFN}\left(Z_{l}^{{}^{\prime }}\right)+Z_{l}^{{}^{\prime }}), \end{array}$$
where FFN(·) is a fully connected feed-forward module.

#### 3.2.4. Data-Hungry Architecture

Inductive bias is defined as a set of assumptions on the data distribution and solution space. In convolutional networks, the inductive bias is inherited and is manifested by the locality and translation invariance. Recurrent networks carry the inductive biases of temporal invariance and locality via their Markovian structure [56]. Transformers have less image-specific inductive bias. They make few assumptions about how the data are structured. This makes the Transformer a universal and flexible architecture, but also prone to overfitting when the data are limited. A possible option to alleviate this issue is to introduce inductive bias into the model by pre-training Transformer models on large datasets. When pre-trained at a sufficient scale, Transformers achieve excellent results on tasks with fewer data. for instance, ViT is pre-trained with a large-scale private dataset called JFT-300M [57]. It manages to achieve similar or even superior results on multiple image recognition benchmarks, such as ImageNet [58] and CIFAR-100 [59], as compared with the most prevailing CNN methods.

#### 3.2.5. Combining Transformers and CNNs in Vision

Both architectures work and learn in different ways, but have the same goal in mind. Therefore, we should aim to combine both architectures. In order to improve representation learning, some integrate Transformers into CCNs, such as BoTNet [60] or VTs [61]. Some go the other way around and enhance Transformers with CNNs, such as DeiT [41], ConViT [62], and CeiT [63]. Because convolutions do an excellent job at capturing low-level local features in images, they have been added at the beginning to patchify and tokenize an input image. Examples of these hybrid designs include CvT [64], LocalViT [65], and LeViT [66].

The patchify process in ViT is coarse and neglects the local image information. In addition to the convolution, researchers have introduced locality into Transformer to dynamically attend to the neighbour elements and augment the local extraction ability. This is performed by either employing an attention mechanism, e.g., Swin [67], TNT [68], and VOLO [10], or using convolutions, e.g., CeiT [63].

Other interesting architectures include hierarchical Transformers (T2T-ViT [69], PVT [70]), and deep Transformers, where the model’s depth strengthens its learning capacity [38], e.g., CaiT [71] and DeepViT [72].

A link to the many published Transformer-based methods is provided here https://github.com/liuyang-ict/awesome-visual-Transformers (accessed on 14 September 2022).

### 3.3. Vision Outlooker

The backbone used in our model, Vision Outlooker (VOLO https://github.com/sail-sg/volo (accessed on 29 September 2022)) [10], proposes an outlook  attention that attends the neighbouring elements to focus on the finer-level features. Similar to patchwise dynamic convolution and involutions [73], VOLO does this by using three operations: unfold, linear weights’ attention, and refold. The dominance of VOLO reflects that the locality is indispensable for Transformer.

The backbone consists of four outlook attention layers, one downsampling operation, followed by three Transformer blocks consisting of various self-attention layers and, finally, two class attention layers. The [cls] token is inserted before the class attention layers. The implementation of VOLO is based on the LV-ViT [74] and the CaiT [71] models and achieves SOTA results in image classification without using any external data.

#### 3.3.1. Outlook Attention

In the core of VOLO sits the outlook attention (OA). for each spatial location (i,j), the outlook attention calculates the similarity between it and all neighbouring features in a local window of size K×K centred on (i,j). The architecture is depicted in Figure 6.

Given an input $X\in R^{H\times W\times C}$, each *C*-dim feature block is projected using two linear layers of weights:$W_{A}\in R^{C\times K^{4}}$ into outlook weights $A\in R^{H\times W\times K^{4}}$;$W_{V}\in R^{C\times C}$ into value representations $V\in R^{H\times W\times C}$.

Let the values within a local window centred at (i,j) be denoted as $V_{\Delta_{i,j}}\in R^{C\times K^{2}}$, where: (3)$$V_{\Delta_{i,j}}=\left\{V_{i+p-\left\lfloor \frac{K}{2}\right\rfloor ,j+q-\left\lfloor \frac{K}{2}\right\rfloor }\right\},0\leq p,q<K.$$

The outlook weight $A_{i,j}$ at location (i,j) is reshaped into ${\hat{A}}_{i,j}\in R^{K^{2}\times K^{2}}$, followed by a SoftMax function, resulting into the attention weight at (i,j). Using a simple matrix multiplication, the weighted average, referred to as value projection procedure, is calculated as
(4)$$Y_{i,j}=\sum_{0\leq m,n<K}Y_{\Delta_{i+m-\left\lfloor \frac{K}{2}\right\rfloor ,j+n-\left\lfloor \frac{K}{2}\right\rfloor }}^{i,j}.$$

The outlook attention is similar to a patchwise dynamic convolution, involution, where the attention weights are predicted by the central feature (performed within local windows) and then folded back (reshaping operation) into feature maps. The self-attention, on the other hand, is calculated using query–key matrix multiplications. Similar to Equation (Equation 2), each Outlooker layer is written as
(5)$$\begin{array}{cc} \tilde{X} & =\mathrm{OA}(\mathrm{LN}(X))+X\\ Z & =\mathrm{MLP}(\mathrm{LN}\left(\tilde{X}\right)+\tilde{X}). \end{array}$$

#### 3.3.2. Class Attention

Introduced by [71], class attention in image Transformers (CaiT) has a deeper and better-optimized Transformer network for image classification. The main difference between CaiT and ViT is the way the [cls] token is compiled. In ViT, the token is attached to the patch embeddings before being fed into the Transformer encoder. In CaiT, the self-attention stage does not take into consideration the class embeddings. This token is only inserted in the class-attention stage, where patch embeddings are frozen, so that the last part of the network is fully devoted to updating the class embedding before being fed to the classifier head.

### 3.4. Transformers in Vehicle Re-Identification

This is a brief literature review on vehicle ReID works using Transformers.

He et al. [47] were the first to introduce pure Transformers in object ReID. Their motivation came from the advantages of pure Transformer-based models being more suitable in CNN-based ReID for the following reasons:Multi-head attention modules are able to capture long-range dependencies and push the models to capture more discriminative parts compared to CNN-based methods;Transformers are able to preserve detailed and discriminative information because they do not use convolution and downsampling operators.

The vehicle images are resized to 256×256 and then split into overlapping patches via a sliding window. The patches are fed into a series of Transformer layers without a single downsampling operation to capture fine-grained information of the image’s object. The authors designed two modules to enhance the robust feature learning, a jigsaw patch module (JPM) and side information embedding (SIE). In re-identification, an object might be partly occluded, leading to only a fragment being visible. Transformer, however, uses the information from the entire image. Hence, the authors proposed a JPM to address this issues. The JPM shuffles the overlapping patch embeddings and regroups them into different parts, helping to improve the robustness of the ReID model. Additionally, an SIE was proposed to incorporate non-visual information, e.g., cameras or viewpoints, to tackle issues due to scene bias. The camera and viewpoint labels are encoded into 1D embeddings, which are then fused with the visual features as positional embeddings. The proposed models achieve state-of-the-art performances on object re-ID, including person (e.g., Market1501 [75], DukeMTMC [76]) and vehicle (e.g., VeRi-776 [3], VehicleID [25]) ReID.

With the aim of incorporating local information, DCAL [77] couples self-attention with a cross-attention module between local query and global key–value vectors. In fact, in self-attention, all the query vectors interact with the key–value vectors, meaning that each query is treated equally to compute the global attention scores. In the proposed cross-attention, only a subset of query vectors interacts with the key–value vectors, which is able to mine discriminative local information in order to facilitate the learning of subtle features. QSA [78] uses ViT as the backbone, and a quadratic  split  architecture to learn global and local features. An input image is split into global parts, then each global part is then split into local  parts, before being aggregated to enhance the representation ability. Graph interactive Transformer (GiT) [79] extracts local features within patches using a local correlation graph (LCG) module and global features among patches using a Transformer.

Other works enhanced CNNs using Transformers. TANet [80] proposes an attention-based CNN to explore long-range dependencies. The method is composed of three branches: (1) a global branch, to extract global features defining the image-level structures, e.g., rear, front, or lights, (2) a side branch, to identify auxiliary side attribute features that are invariant to viewpoints, e.g., colour or car type, and (3) an independent attention branch, able to capture more detailed features. Using a CNN-based backbone, MsKAT [81] consists of a ResNet-50 backbone coupled with a knowledge-aware Transformer.

In the AI City Challenge 5 [21], DMT [82] used TransReID as the backbone to extract global features via the [cls] token. Due to computational resources and the non-availability of side information, the JPM and SIE modules were removed. The authors achieved a 74.45% mAP score for Track 2. Other works can be found in Table 4.

As we notice, papers that achieved good results either used Transformer-enhanced CNNs or included additional information. Codes are only available for TransReID https://github.com/heshuting555/TransReID (accessed on 1 September 2022) [47] and DMT https://github.com/michuanhaohao/AICITY2021_Track2_DMT (accessed on 1 September 2022) [82]. We show that it is possible to achieve state-of-the-art results by only using images as the input. Furthermore, we provide a well-documented code.

### 3.5. Designing Your Loss Function

Apart from model designs, loss functions play key roles in training a ReID network. In accordance with the loss function, ReID models can be categorized into two main genres: classification loss for verification tasks and metric loss for ranking tasks.

#### 3.5.1. Classification Loss

The SoftMax function [84,85] and the cross-entropy [86] are combined together into the cross-entropy loss, or SoftMax loss. The latter is sometimes referred to as classification loss in classification problems or as ID loss when applied in ReID [87]. Let *y* be the true ID of an image and $p_{i}$ the ID prediction logits of class *i*. The ID loss is computed as:(6)$$L_{id}=\sum_{i=1}^{N}-q_{i}\log\left(p_{i}\right)\left\{\begin{array}{c} q_{i}=0,y\neq i\\ q_{i}=1,y=i \end{array}\right.$$

ID loss requires an extra fully connected (FC) layer to predict the logits of IDs in the training stage. Furthermore, it cannot solve the problem of large intra-class similarities and small inter-class differences. Some improved methods such as large margin (L)-SoftMax [88], angular (A)-SoftMax [89], and virtual SoftMax [90] have been proposed. As the category in closed vehicle ReID is fixed, the classification loss is commonly used. However, the category can change based on different vehicle models or different quantities of vehicles over time. A model trained using only the ID loss leads to poor generalization ability. Therefore, Hermans et al. [91] emphasized that using the triplet loss [92] can lead to better performances than the ID loss.

#### 3.5.2. Metric Loss

Among some common metric losses are the triplet loss [91], the contrastive loss [27], the quadruplet loss [93], the circle loss [94], and the centre loss [95]. Our proposed V${}^{2}$ReID uses the triplet and centre loss for training.

The triplet loss regards the ReID problem as a ranking problem. Models based on the triplet loss take three images (triplet sample) as the input: one anchor image $x_{a}$, one image with the same ID as the anchor $x_{p}$ (positive), and one image with a different ID from the anchor $x_{n}$ (negative). The margin α is enforced to ensure distance between positive and negative pairs. The triplet is then denoted as $t=(x_{a},x_{p},x_{n})$, and the triplet loss function is formulated as
(7)$$L_{tri}\left(x_{a},x_{p},x_{n}\right)=max\left(\alpha +d_{ap}-d_{an},0\right),$$
where d(.) measures the Euclidean distance between two samples and α is the margin threshold that is enforced between positive and negative pairs. The selection of samples for the triplet loss function is important for the accuracy of the model. When training the model, there should be both an easy pair and a difficult pair. The easy pair should have a small distance or a slight change between the two images. Changes can be in the rotation of an image or other small changes. The hard pair would be a more significant change in either clothing, surroundings, lighting, or other drastic changes. Doing this can improve the accuracy of the triplet loss function. When incorporating the triplet loss, the data need to be sampled in a specific way. A sampler indicates how the data should be loaded. As for the triplet loss, we need positive and negative images, and we need to make sure that during data loading, we have *k* instances for each identity per batch.

The triplet loss only considers the relative distance between $d_{ap}$ and $d_{an}$ and ignores the absolute distance. The centre loss can be used to minimize the intra-class distance in order to increase the intra-class compactness. This improves the distinguishability between features. Let $c_{y_{i}}\in R^{d}$ be the $y_{i}^{th}$ class centre of deep features. The centre loss is formulated as
(8)$$L_{cen}=\frac{1}{2}\sum_{i=1}^{m}\left|\right|x_{i}-c_{y_{i}}{\left|\right|}_{2}^{2}.$$

Ideally, $c_{y_{i}}$ should be updated as the deep features change.

#### 3.5.3. Combining Classification and Metric Loss

Unifying the triplet loss and the classification loss improves the model performance. Most works use that combination formulated as: (9)$$L=\lambda_{id}L_{id}+\lambda_{tri}L_{tri}.$$

Examples of works include SCAN [17], TransReID [47], GiT [79], DMT [82], QSA [78], or DCAL [77].

Conventionally, the weights of the ID and metric loss are set to 1:1. In practice, there is an imbalance between both losses, and changing the ratio can improve the performance [23]. The authors showed that using a 0.5:0.5 ratio can improve the mAP score of VOC-ReID [96] by 3.5%. They proposed a momentum adaptive loss weight (MALW), which automatically updates the loss weights according the the statistical characteristics of the loss values, and combined the CE loss and the supervised contrastive loss [97], achieving an 87.1% mAP on VeRi. Reference [95] adopted the joint supervision of SoftMax loss and centre loss to train their CNN for discriminative learning. The formulation is given by
(10)$$L=L_{id}+\lambda L_{cen}.$$

Luo at el. [98] went a step further and included three losses: (11)$$L=L_{id}+L_{tri}+\lambda_{cen}L_{cen}$$
where β is the balanced weight of the centre loss, set to 0.0005.

### 3.6. Techniques to Improve Your Re-Identification Model

Here, we summarize two techniques from [98], who proposed a list of training tricks to enhance the ReID model.

#### 3.6.1. Batch Normalization Neck

Most ReID works combine the ID loss and the triplet loss to learn more discriminative features. It should be noted, however, that classification and metric losses are inconsistent in the same embedding space. ID loss constructs hyperplanes to separate the embedding space into different subspaces, making the cosine distance more suitable. Triplet loss, on the other hand, tries to optimize the Euclidean distance, as it tries to draw closer similar objects (decrease intra-class distance) while pushing away different objects (increase inter-class distance) in the Euclidean space. When using both losses simultaneously, their goals are not consistent, and it can even lead to one loss being reduced while the other one is increased.

In standard baselines, the ID loss and triplet loss are based on the same features, meaning that features *f* are used to calculate both losses (see Figure 7). Luo et al. [98] proposed the batch normalization neck (BNNeck), which adds a batch normalization layer after the features (see Figure 7). The PyTorch-style command for adding the BNNeck is given in Algorithm 2.
**Algorithm 2: **
PyTorch-style command for BNNeck.
# x = output of networkglobal_feat = xif neck == ’no’:    feat = global_featelse:    feat = nn.BatchNorm1d(global_feat)x_cls = nn.Linear(feat)# return: cls for ID, global_feat for triplet lossreturn x_cls, global_feat

#### 3.6.2. Label Smoothing

Label smoothing is a regularization technique that introduces noise for the labels. This accounts for the fact that datasets may have mistakes in them, so maximizing the likelihood of logp(y|x) directly can be harmful. Assume for a small constant ϵ that the training set label *y* is correct with probability 1−ϵ and incorrect otherwise.

Szegedy at al. [99] proposed an LS mechanism to regularize the classifier layer, to alleviate overfitting for a classification task. This mechanism assumes that there may be errors in the label during training to prevent overfitting. The difference is how $q_{i}$ is calculated in the ID loss (Equation (Equation 6)):(12)$$q_{i}=\left\{\begin{array}{cc} 1-\frac{N-1}{N}\epsilon & \mathrm{if}\;i=y\\ \frac{\epsilon }{N} & \mathrm{otherwise},\; \end{array}\right.$$
where i∈1,2,…,N represents the sample category, *y* represents the truth ID label, ϵ is a constant to encourage the model to be less confident in the training set, i.e., the degree to which the model does not trust the training set, and was set to 0.1 in [23,100].

### 3.7. V${}^{2}$ReID Architecture

Taking everything into account, we present our final architecture of V^2^ReID using VOLO as the backbone, as outlined in Figure 8. The steps are as follows:Preparing  the  input data  (1)–(2): The model accepts as input mini-batches of three-channel RGB images of shape (*H* × *W* × *C*), where *H* and *W* are the height and width. All the images then go through data augmentation such as normalization, resizing, padding, flipping, etc. After the data transform, the images are split into non-overlapping or overlapping patches. While ViT uses one convolutional layer for non-overlapping patch embedding, VOLO uses four layers. Besides the number of layers, there is also a difference in the size of the patches. In order to encode expressive finer-level features, VOLO changes the patch size (*P*) from 16×16 to 8×8. The total number of patches is then $N=HW/P^{2}$.VOLO Backbone  (3)–(7): VOLO comprises Outlooker (3), Transformer (5) and Class Attention (7) blocks. A [cls] token (6) is added before the class attention layers (7). Depending on the model variant (D1–D5), the number of layers per block differs. After the patch embeddings (2) go through the Outlooker block (3), the tokens are downsampled (4). Positional encoding is then added, and the tokens are fed into the Transformer blocks.Classifying the vehicle (8)–(10): The output features (8) are run through the classifier heads (10), consisting of different losses. Optionally, when using the BNNeck, it is inserted in (9).

## 4. Datasets and Evaluation

### 4.1. Datasets

VeRi-776: VeRi-776 [101] is an extension of the VeRi dataset introduced in [3] https://vehiclereid.github.io/VeRi/ (accessed on 14 May 2022). VeRi is a large-scale benchmark dataset for vehicle ReID in the real-world urban surveillance scenario featuring labels of bounding boxes, types, colours, and brands. While the initial dataset contains about 40,000 images of 619 vehicles captured by 20 surveillance cameras, VeRi-776 contains over 50,000 images of 776 vehicles. Furthermore, the dataset includes spatiotemporal information, as well as cross-camera relations, license plate annotation, and eight different views, making the dataset scalable enough for vehicle ReID. VeRi-776 is divided into a training subset containing 37,746 images of 576 subjects and a testing subset including a probe subset of 1678 images of 200 subjects and a gallery subset of 11,579 images of the same 200 subjects.

Vehicle-ID: Vehicle-ID [25] is a surveillance dataset, containing 26,267 vehicles and 221,763 images in total. The camera IDs are not available. Each vehicle only has the front and/or back viewpoint images (two views). The training set includes 110,178 images of 13,134 vehicles, and the testing set consists of three testing subsets at different scales, i.e., Test-800 (S), Test-1600 (M) and Test-2400 (L). As our paper presents details on how to train and improve your model, we do not present any results on the Vehicle-ID dataset.

An extensive list of vehicle ReID benchmarks can be found via https://github.com/bismex/Awesome-vehicle-re-identification (accessed on 19 August 2022).

### 4.2. Evaluation

In closed-set ReID, the most common type of comparison found in the literature between each model is cumulative matching characteristics (CMCs), and mean average precision (mAP).

CMCs: Cumulative matching characteristics are used to assess the accuracy of a model, which produce an ordered list of possible matches. Referred to also as the rank-*k* matching accuracy, CMCs indicate the probability that a query identity appears in the top-*k* ranked retrieved results. They treat the re-identification as a ranking problem, where given one or one set of query images, the candidate images in the gallery are ranked according to their similarities to the query. For each query, the cumulative match score is calculated based on whether there is a correct result within the first *R* columns, with *R* being the rank. Summing these scores gives us the cumulative matching characteristics. for instance, if the rank-10 has an accuracy of 50%, it means that the correct match occurs somewhere in the top 10, 50% of the time. The CMCs’ top-k accuracy is formulated as: (13)$$Acc_{k}=\left\{\begin{array}{cc} 1, & \mathrm{if}\;\mathrm{query}\;\mathrm{ID}\;\mathrm{is}\;\mathrm{in}\;\mathrm{top}-k\;\mathrm{gallery}\;\mathrm{samples}\\ 0, & \mathrm{otherwise}. \end{array}\right.$$

mAP: The mean average precision has been widely used in object detection and image retrieval tasks, especially in ReID. Compared to CMCs, the mAP measures the retrieval performance with multiple ground truths. While the average precision (AP) measures how well the model judges the results on a single query image, the mean average precision (mAP) measures how well the model judges the results on all query images. The mAP is the average of all the APs, and both can be calculated as follows: (14)$$\begin{array}{cc} AP & =\frac{\sum_{k=1}^{n}p\left(k\right)g\left(k\right)}{N_{g}}\\ mAP & =\frac{\sum_{q=1}^{Q}AP\left(q\right)}{Q}, \end{array}$$
where *n* is the number of test images and $N_{g}$ is the number of ground truth images, p(k) is the precision at the *k*-th position, and g(k) represents the indicator function, where the value is 1 if the *k*-th result is correct, else 0. The mean average precision (mAP) is calculated as follows, where *Q* is the number of images queried.

Example: Given an example of queries and the returned ranked gallery samples (see Figure 9), here is a detailed example of three queries, where the CMCs are 1 for all rank lists, while the APs are 1, 1 and 0.7. The calculations for each query are: (15)g1=10000,p1=1112131415,N1=1g2=10000,p2=1122232425,N2=2g3=10001,p3=1112131425,N3=2⇒AP1=1AP2=1AP3=710

## 5. Experiments and Results

The original VOLO code and the pre-trained models on ImageNet-1k [58] are available on GitHub https://github.com/sail-sg/volo (accessed on 14 May 2022). The token labelling part inspired by LV-ViT [74] is not used in V${}^{2}$ReID. The following paragraphs summarize our experiments with a discussion of the results.

### 5.1. Implementation Details

The proposed method was trained in Pytorch. We ran our experiments on one NVIDIA A100 PCIe with 80 GB VRAM.

#### 5.1.1. Data Preparation

All models accept as the input mini-batches of 3-channel RGB images of shape (*H* ×*W*× *C*), where *H* and *W* were set to 224, unless mentioned otherwise. All the images were normalized using ImageNet’s mean and standard deviation. Besides normalizing the input data, we also used other data augmentation settings, such as padding, horizontal flipping, etc. https://pytorch.org/vision/stable/transforms.html (accessed on 18 May 2022). Figure 10 illustrates the transforms using exaggerated values.

#### 5.1.2. Experimental Protocols

In the following paragraphs, the performance changes using various settings for a chosen hyperparameter are analysed. More specifically, we compared different models based on the pre-training (Section 5.2.2), the loss function (Section 5.2.3), and the learning rate (Section 5.2.4). Once we found the best model, we pushed it further by testing it using different optimizers (Section 5.2.5) and VOLO variants (Section 5.2.6). We detected some training instability and aimed to solve this using learning rate schedulers (Section 5.2.7). For each table, the best mAP and R-1 scores are highlighted. The protocols for how to read our results are in Table 5.

### 5.2. Results

#### 5.2.1. Baseline Model

The baseline model was tuned with the settings indicated in Table 6. The values were inspired by the original VOLO [10] and TransReID [47] papers. While VOLO uses AdamW [102] as the optimizer, V${}^{2}$ReID adopted the SGD optimizer in those experiments with a warm-up strategy to bootstrap the network [103]. The baseline model was trained using the ID loss. Given the base learning rate (LR${}_{\mathrm{base}}$), we spent 10 epochs linearly increasing LR × 10${}^{-1}$→ LR. Unless mentioned otherwise, cosine annealing was used as the learning rate scheduler [47,80,96].

#### 5.2.2. The Importance of Pre-Training

The best way to use models based on Transformers is to pre-train them on a large dataset before fine-tuning them for a specific task. The pre-trained models can be downloaded from the VOLO GitHub.

In Table 7, the different experiment IDs indicate the same model (based on loss functions, neck settings, learning rates, and weight decay values), and we compared the performances of pre-training vs. from-scratch training.

Except for Experiment 1, the pre-trained model always performed better. When inspecting the models trained from-scratch, Experiment 5 performed best with a 59.71% mAP and 89.39% R-1. On the other hand, using a pre-trained model, Experiment 4 achieved the highest scores, 78.03% mAP and 96.24% R-1. Fine-tuning the model can boost the mAP between 17 and 21%. For the rest of the paper, only pre-trained models were used.

#### 5.2.3. The Importance of the Loss Function

The total loss function used is:$$L_{tot}=\lambda_{ID}L_{ID}+\lambda_{tri}L_{tri}+\lambda_{cen}L_{cen},$$
where $L_{ID}$ is the cross-entropy loss, $L_{tri}$ the triplet loss, and $L_{cen}$ the centre loss. Following common practices found in the literature, the weights were set to $\lambda_{ID}=\lambda_{tri}=1$ and $\lambda_{cen}=0.0005$. Referring to Figure 8, the features in Step 8 were used to compute $L_{tri}$ and $L_{cen}$, while the features after the classifier head in Step 10 were used to compute $L_{ID}$. We compared the models (trained with/without BNNeck and with different loss functions) using the same learning rates ($1.0\times 10^{-3}$, $2.0\times 10^{-3}$, and $1.5\times 10^{-2}$). Table 8 summarizes different scores depending on the loss functions.

The best results were achieved when using the three losses, without the BNNeck and a learning rate of $2.0\times 10^{-3}$. Experiments 2 and 3 showed that combining the ID loss with the triplet loss and the centre loss did not deal well with a bigger learning rate of $1.5\times 10^{-2}$. The latter was preferred by Experiment 4, using the BNNeck. Interested in the training behaviour, we plot Figure 11, which shows the loss and mAP per epoch for different loss functions and learning rates. Training using a BNNeck (in red) converged much faster, compared to its counterparts.

Finally, we replaced the batch normalization neck with a layer normalization neck (LNNeck); see Table 9. The model was tested using four different learning rates, and it performed best for a base learning rate of $1.0\times 10^{-3}$.

As the unified ID, triplet, and centre loss performed best, we kept that loss for the rest of the paper. We continued to experiment with and without the BNNeck.

#### 5.2.4. The Importance of the Learning Rate

“The learning rate is perhaps the most important hyperparameter. If you have time to tune only one hyperparameter, tune the learning rate.”[104]

If the learning rate is too large, the optimizer diverges; if it is too small, then training takes too long or we end up with a sub-optimal result. The optimal learning rate is dependent on the topology of the loss function, which is in turn dependent on both the model architecture and the dataset. We experimented on different learning rates to find the optimal rate for our model. Table 10 summarizes the results based the same loss functions using different learning rates. For the same loss functions and a BNNeck, Figure 12 shows different scores.

The model without a BNNeck was able to achieve an mAP score of 78.02% for a learning rate of $2.0\times 10^{-3}$ and an R-1 score of 96.90% for a learning rate of $1.9\times 10^{-3}$. When using a BNNeck, the best performance we found was 77.41% mAP for a learning rate of $1.5\times 10^{-2}$ and 96.72% R-1 for a learning rate of $9.0\times 10^{-3}$.

In the next subsections, we used a learning rate of $1.5\times 10^{-3}$ with a BNNeck and $2.0\times 10^{-3}$ without a BNNeck.

#### 5.2.5. Using Different Optimizers

Our next step was to test different optimizers. We adapted the standard SGD as in [47] and kept the same learning rate to test the models using AdamW and RMSProp. The loss function was the unified ID, triplet, and centre loss, without the BNNeck. Table 11 gives the mAP and R-1 scores for various learning rates. For the learning rate that we tested, SGD achieved the best results. Both AdamW and RMSProp performed better using a smaller learning rate.

#### 5.2.6. Going Deeper

We are interested in whether the depth of the model can enhance the performance. Going deeper means, for a given training hardware, more parameters, longer runtimes, and smaller batch sizes.

First, we tested the models using different batch sizes; see Table 12. In terms of the mAP scores, using a bigger batch size produced better results, in reference to Experiments 1, 3, and 4. In Experiment 3, using a bigger batch size boosted the mAP by 3.42%. Unfortunately, we could not test bigger batch sizes with the larger models (D3–D5) because of the GPU being limited to 80 GB.

The next step was to test different model variants (VOLO D1-D5); see Table 13. All the hyperparameters remained the same for the variants, and we used the three losses, the BNNeck, and a learning rate of 0.0150. The batch sizes differed to accommodate the storage. Using VOLO-D5, we had an increase of 2.89% in mAP. We achieved, to our knowledge, the best results in vehicle ReID using a Transformer-based architecture taking as the input only visual cues provided by the input images.

The loss and mAP% evolution during the learning process are shown in Figure 13. Interestingly, VOLO-D3 presented a sudden spike in the loss/trough in the mAP score, at Epoch 198. This observed behaviour was not detected when training the model without using the BNNeck.

Interested in the learning behaviour of VOLO-D3 using a BNNeck, we first changed the learning rate by small increments; see Figure 14. The tested learning rates were 0.015000 (orange), 0.0150001 (blue), 0.015001 (green), and 0.015010 (red). We concluded that using a smaller increment of $1.0\times 10^{-6}$ and $1.0\times 10^{-7}$ can render the training more stable. The best results were achieved with a learning rate of 0.015001.

#### 5.2.7. The Importance of the LR Scheduler

Finally, we were curious to know whether changing the settings of the cosine annealing [102,105] can render the training more stable. Using cosine annealing for each batch iteration *t*, the learning rate $\eta_{t}$ was decayed within the *i*-th run as follows [102]: (16)$$\eta_{t}=\eta_{min}^{i}+\frac{1}{2}\left(\eta_{max}^{i}-\eta_{min}^{i}\right)\left(1+\cos\left(\frac{T_{cur}}{T_{i}}\pi \right)\right),$$
with $\eta_{min}^{i}$ and $\eta_{max}^{i}$ the ranges for the learning rate, $T_{cur}$ the number of epochs that were performed since the last restart, and *T* the number of total epochs. Figure 15 visualizes how the learning rate evolved using different settings. For more information, please refer to https://timm.fast.ai/SGDR (accessed on 28 August 2022).

We tested VOLO-D3 with a learning rate of $1.5\times 10^{-2}$ by changing the settings of the cosine decay:Linear warm-up: Figure 16 visualizes how the loss and mAP varied depending on the number of warm-up epochs. Without using any warm-up (blue), the spike in the loss was deeper and it took the model longer to recover from it. When using a warm-up of 50 epochs (green), the spike was narrower. Finally, testing using 75 warm-up epochs, there was no spike during the training.Number of restart epochs: Figure 17 shows the evolution of the learning rate using different numbers of restart epochs (140, 150, 190) and decay rates (0.1 or 0.8). The decay rate is the value by which, at every restart, the learning rate is decayed by, using the following multiplication: LR × decay_rate. When using 150 restart epochs with a decay rate of 0.8 (orange), the mAP score dipped, but recovered quickly and achieved a higher score compared to the two others. When restarting with 140 epochs (blue) or 190 epochs (green), both with a decay rate of 0.1, there was no dip in the mAP during training; however, the resulting values were lower.

#### 5.2.8. Visualization of the Ranking List

Finally, we are interested in visualizing the discriminatory ability of the final model, which achieved an 80.3% mAP. Given a query image (yellow) in Figure 18, we retrieved the top-10 ranked results from the gallery that the model deemed to be the most similar to the query. Five of the most-interesting outputs are shown in order. The images with a red border are incorrect matches, while the ones with a green border correspond to the correct vehicle ID. Some observations are as follows:1.Our model was able to identify the correct type of vehicle (model, colour).2.The same vehicle can be identified from different angles/perspectives (see the first and last rows).3.Occlusion and illumination can interfere with the model’s performance (see the 1st and 2nd rows).4.Using information on the background and the timestamp would enhance our model’s predictive ability. Looking at the third row, the retrieved vehicle was very similar to the query vehicle. However, when looking at the background, there was information (black car) that was not detected. As for the fourth row, there was no red writing on the wrong match; furthermore, that truck carried more sand than the truck from the query.5.Overall, the model was highly accurate at predicting the correct matches. As a human, we would have to look more than twice to grasp the tiny differences between the query and the retrieved gallery images.

## 6. Conclusions

This paper had two main goals: (1) implementing a novel vehicle re-identification model based on Vision Outlooker and (2) documenting the process in an approachable way.

We implemented V^2^ReID using Vision Outlooker as the backbone and showed that the outlook attention was beneficial to the vehicle recognition task. The hyperparameters such as pre-training, learning rate, loss function, optimizer, VOLO variants, and learning rate schedulers were analysed in depth in order to understand how each of them can impact the performance of our model. It uses less parameters compared to other approaches and was thus able to infer results faster. V^2^ReID achieved successfully an 80.30% mAP and 97.13% R-1, by using only the VeRi-776 dataset as the input, without any other additional information. All this process was documented in an easy-to-understand way, as this is rarely available in the literature. Our paper can be used as a walk-through for anyone that is getting started in this field by providing the most details and by grouping various types of information into one single paper.

The proposed V^2^ReID serves as a baseline for future object re-identification and multi-camera multi-target re-identification applications. Further study includes (1) testing on other hyperparameters such as image size, patch size, overlapping patches, values of $\lambda_{ID},\;\lambda_{tri},\;\lambda_{cen}$, etc., (2) enhancing the performance by adding additional information such as the timestamp, background and vehicle colour detection, etc., (3) designing a new loss function that is consistent in the same embedding space, and finally, (4) including synthetic data in order to overcome the lack of data and to deal with inconsistency in the distribution of different data sources.

## Figures and Tables

**Figure 1 sensors-22-08651-f001:**
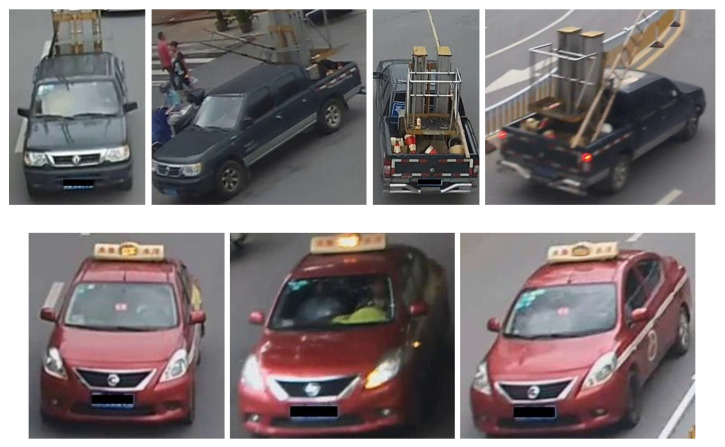
(**Top row**) Large intra-class differences, i.e., same vehicle looking different from distinct perspectives; (**bottom row**) small inter-class similarities, i.e., different vehicles looking very similar.

**Figure 2 sensors-22-08651-f002:**
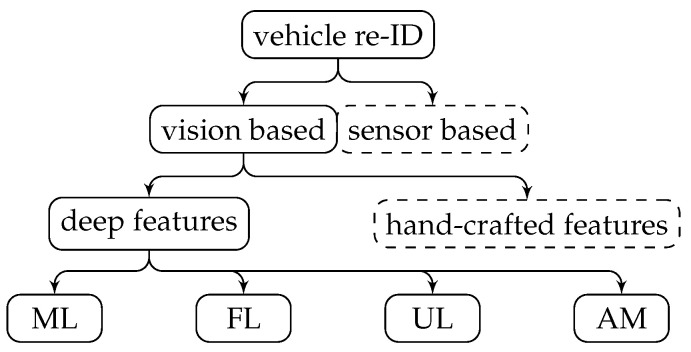
Categories of vehicle ReID methods. Dashed boxes represent methods that are not detailed.

**Figure 3 sensors-22-08651-f003:**
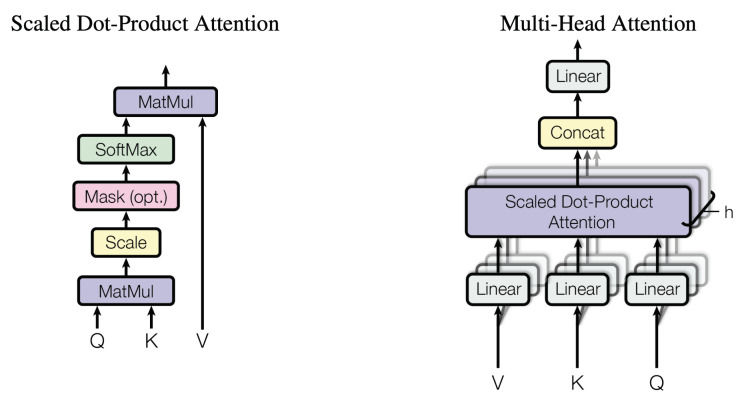
(**Left**) Scaled dot product attention; (**right**) multi-head attention [39].

**Figure 4 sensors-22-08651-f004:**
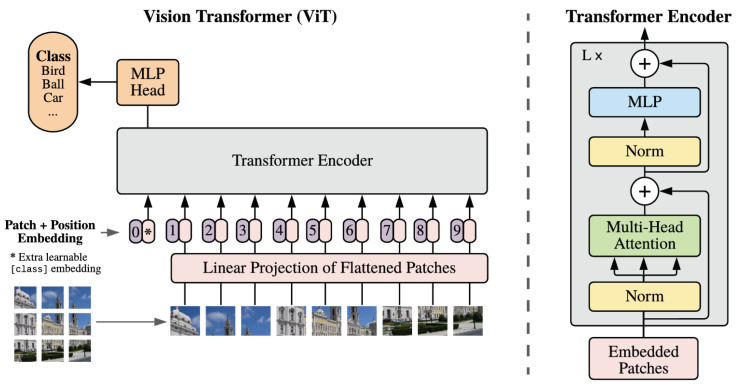
ViT overview [40]: (**Left**) an image is split into patches, each patch is linearly embedded and fed into the Transformer Encoder; (**right**) the building blocks of the Transformer Encoder.

**Figure 5 sensors-22-08651-f005:**
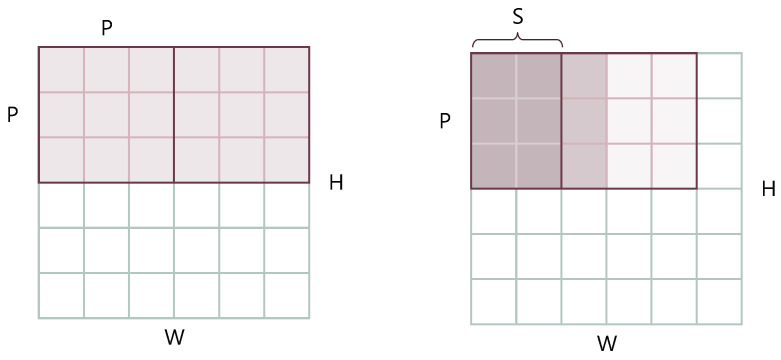
Non-overlapping patches (**left**) vs. overlapping patches (**right**).

**Figure 6 sensors-22-08651-f006:**
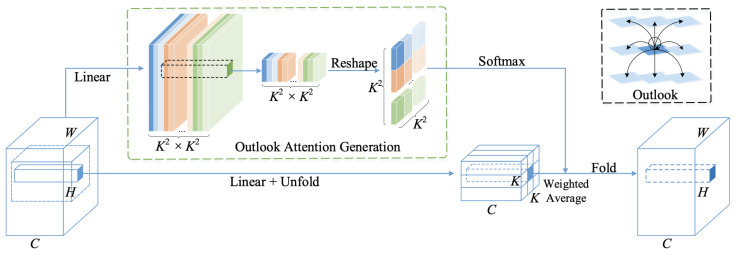
Detailed illustration of the outlook attention from [10].

**Figure 7 sensors-22-08651-f007:**
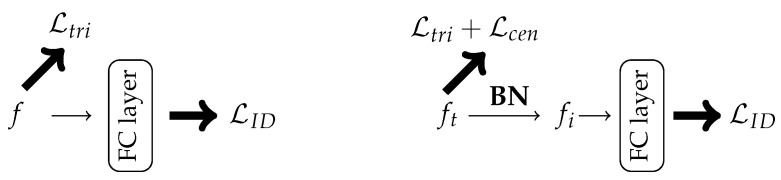
The pipeline of the standard baseline (**right**) and the proposed BNNeck [98].

**Figure 8 sensors-22-08651-f008:**
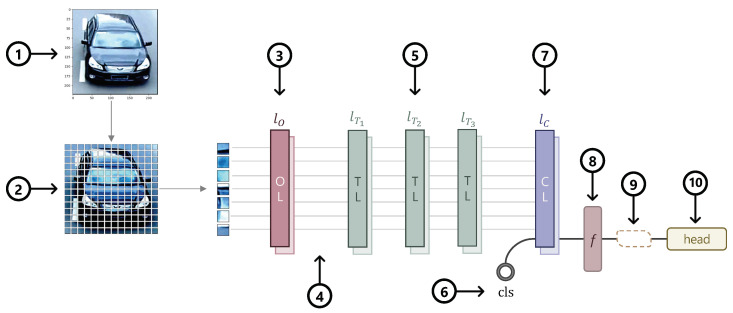
Illustration of V^2^ReID using Vision Outlooker as the backbone. The numbers denote each step: from splitting the input image into fixed-size patches, to feeding the patches in VOLO, to classifying the input image.

**Figure 9 sensors-22-08651-f009:**
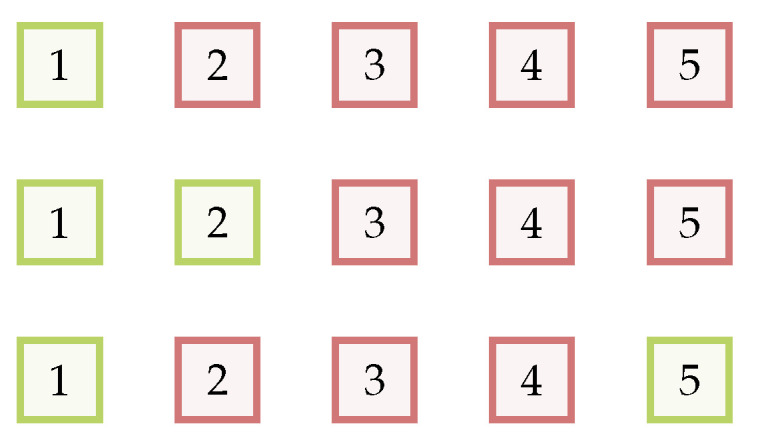
Example of rank lists of queries and the returned ranked gallery sample. Green means a correct match, and red means the wrong matching. For all rank lists, the CMCs are 1 while the AP are 1, 1 and 0.7.

**Figure 10 sensors-22-08651-f010:**
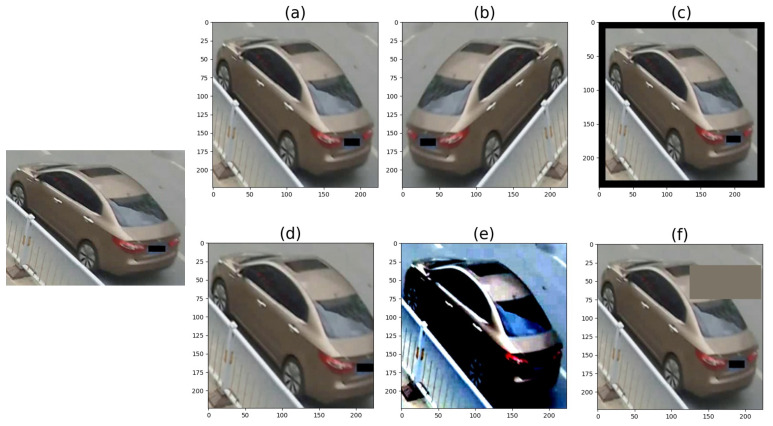
Samples of data augmentation methods: input (**left**), resizing (**a**), horizontal flipping (**b**), padding (**c**), random cropping and resizing (**d**), normalizing (**e**), and random erasing (**f**).

**Figure 11 sensors-22-08651-f011:**
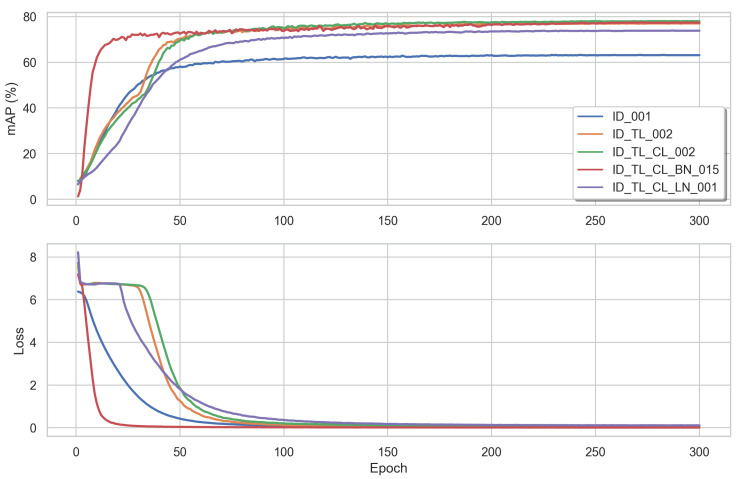
The mAP score (%) and training loss per epoch using different loss functions and learning rates: $L_{ID}$ and LR = $1.0\times 10^{-3}$ (blue), $L_{ID},L_{tri}$ and LR = $2.0\times 10^{-3}$ (yellow), $L_{ID},L_{tri},L_{cen}$ and LR = $2.0\times 10^{-3}$ (green), $L_{ID},L_{tri},L_{cen}$, BNNeck and LR = $2.0\times 10^{-3}$ (red), and LNNeck and LR = $1.0\times 10^{-3}$ (purple).

**Figure 12 sensors-22-08651-f012:**
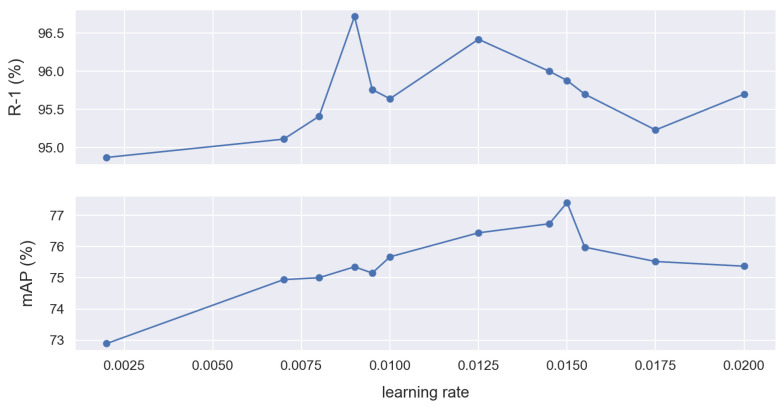
The mAP and R-1 scores in % for different learning rate values using $L_{id},\;L_{tri},\;L_{cen}$ and the BNNeck.

**Figure 13 sensors-22-08651-f013:**
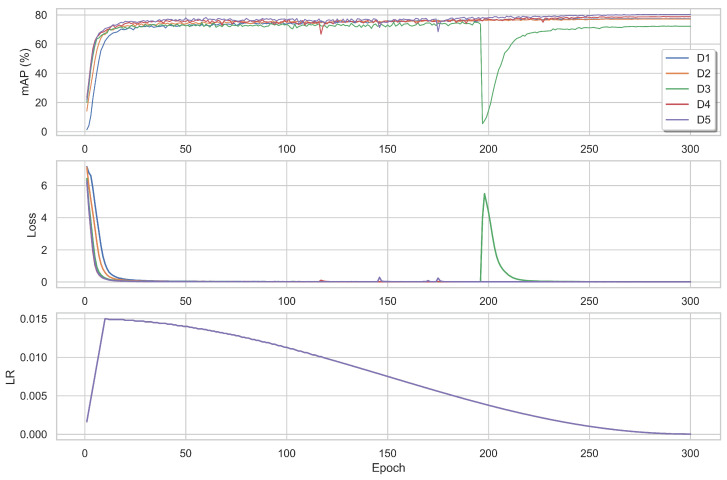
The mAP scores and training loss per epoch for different variants using BNNeck and a base learning rate of 0.0150. The bottom figure shows how the learning rate decays per epoch using the cosine annealing.

**Figure 14 sensors-22-08651-f014:**
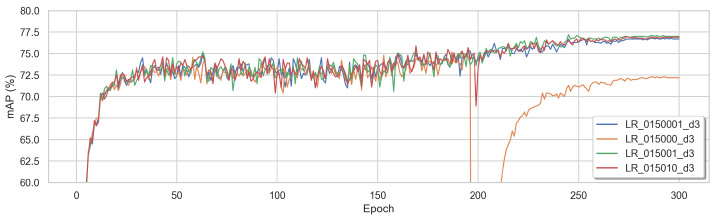
The mAP in % per epoch when training VOLO-D3 using the three losses, BNNeck with different learning rates: 0.015000 (orange), 0.0150001 (blue), 0.015001 (green), and 0.015010 (red).

**Figure 15 sensors-22-08651-f015:**
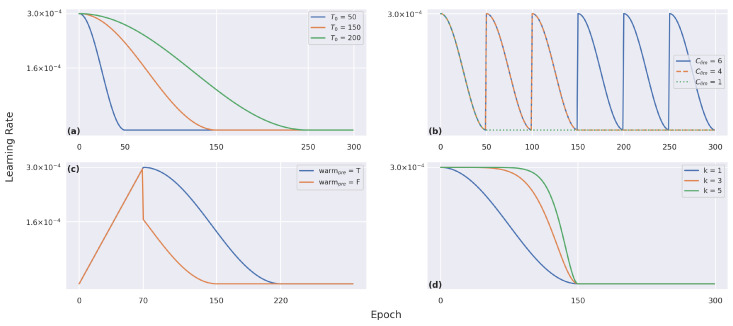
Visualization of the learning rate decay using the cosine annealing decay with a base learning rate of $3.0\times 10^{-4}$, based on (**a**) the initial number of epochs, (**b**) the number of maximum restarts, (**c**) a warm-up of 70 epochs using a pre-fix, and (**d**) the k-decay rate from [105].

**Figure 16 sensors-22-08651-f016:**
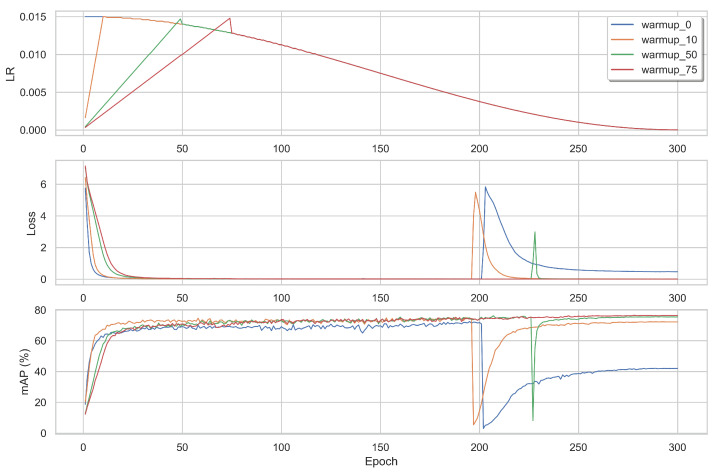
The mAP, loss, and learning rate per epoch when training D3 using the three losses and the BNNeck. The learning rate is linearly warmed up with different numbers of epochs until reaching LR_base_ = 0.0150. The LR is then decayed using cosine annealing for 300 epochs.

**Figure 17 sensors-22-08651-f017:**
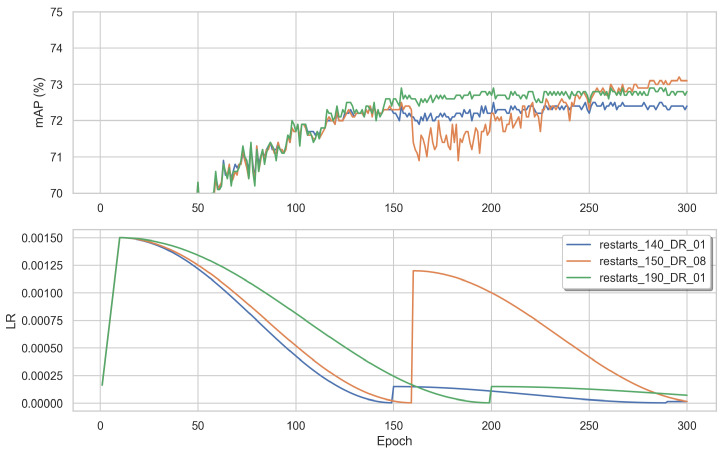
The mAP, loss, and learning rate per epoch when training D3 using the three losses and the BNNeck. The learning rate is linearly warmed up of 10 epochs until reaching LR_base_ = 0.0150. The LR is then decayed using cosine annealing for 300 epochs with different restart values (140, 150, and 190) and decay rates (0.1 and 0.8).

**Figure 18 sensors-22-08651-f018:**
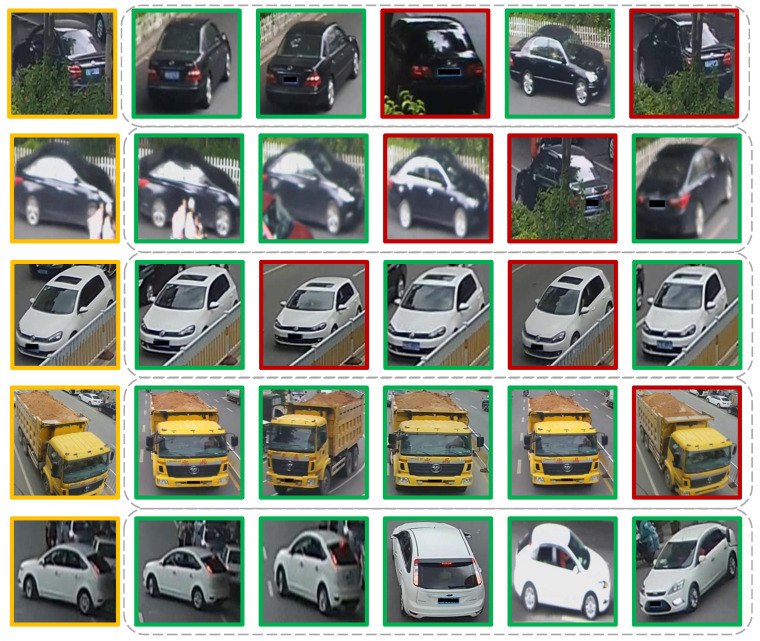
Visualization of five different predicted matches, shown in order from the top-10 ranking list. Given a query (yellow), the model either retrieves a match (in green) or a non-match (red).

**Table 1 sensors-22-08651-t001:** Surveys on vehicle re-identification.

Year	Title	Setting	Based on
2017	Vehicle Re-identification in Camera Networks: A Review and New Perspectives [13]	Closed	sensor, vision
2019	A Survey of Advances in Vision-Based Vehicle Re-identification [14]	Closed	sensor, vision
2019	A Survey of Vehicle Re-Identification Based on Deep Learning [11]	Closed	vision
2021	Trends in Vehicle Re-identification Past, Present, and Future: A Comprehensive Review [5]	Closed	sensor, vision

**Table 2 sensors-22-08651-t002:** Vehicle ReID methods based on deep-features [11].

Method	Description	Advantages	Disadvantages
Local feature (LF)	Focuses on the local areas of vehicles using key point location and region segmentation	Able to capture unique visual cues; can be combined with global features	Extraction of local features is resource intensive
Metric learning (ML)	Focuses on the details of the vehicle by learning the similarity of vehicles	Achieves high accuracy	Needs to design a loss function
Unsupervised learning (UL)	No need for labelled data	Improves the generalization ability; solves the domain shift	Training is unstable
Attention mechanism (AM)	Model learns to identify what areas need to be paid attention to; self-adaptively extracts features	Learns what areas to focus on; extracts features of distinguishing regions	Poor effect when using few labelled data or complex backgrounds

**Table 4 sensors-22-08651-t004:** Summary of some results on vehicle re-identification in a closed-set environment using Transformers on the VeRi-776 [3] and Vehicle-ID [25] datasets. * denotes methods using additional information, e.g., state or attribute information, and ^†^ methods detecting attribute information.

Year	Model	VeRi-776		Vehicle-ID (mAP (%)/R-1 (%))		
		mAP (%)	Rank-1 (%)	S	M	L
2021	TransReID * [47]	82.30	97.10	-	-	-
2021	TransReID (ViT-Base) [47]	78.2	96.5	82.3/96.1	-	-
2021	GiT * [79]	80.34	96.86	84.65/ -	80.52/ -	77.94/ -
2022	VAT * [83]	80.40	97.5	84.50/ -	80.50/ -	78.20/ -
2022	QSA * [78]	82.20	97.30	88.50/98.00	84.70/96.30	80.10/92.10
2022	DCAL [77]	80.20	96.90	-	-	-
2022	MsKAT * [81]	82.00	97.10	86.30/97.40	81.80/95.50	74.90/93.90
2022	TANet ^†^ [80]	80.50	95.4	88.20/82.9	87.0/81.5	85.9/79.6

**Table 5 sensors-22-08651-t005:** Instructions on how to read the different values of the result tables.

Column Name	Values	Comments
ID	natural number	identifier of the experiment
pre-trained	✓	true (pre-trained)
	✗	false (from scratch)
loss	$L_{ID}$	$L_{tot}=1\times L_{ID}$
	$L_{ID},L_{tri}$	$L_{tot}=1\times L_{ID}+1\times L_{tri}$
	$L_{ID},L_{tri},L_{cen}$	$L_{tot}=1\times L_{ID}+1\times L_{tri}+0.0005\times L_{cen}$
BNNeck	✓	using batch normalization neck
	✗	not using batch normalization neck

**Table 6 sensors-22-08651-t006:** Settings of the baseline model. *ss.* refers to the subsections where the hyperparameter is analysed.

Specifications	Value
variant (Section 5.2.6)	VOLO-D1
pre-trained (Section 5.2.2)	false
optimizer (Section 5.2.5)	SGD
momentum	0.9
base learning rate (Section 5.2.4)	$1.6\times 10^{-3}$
weight decay	$5.0\times 10^{-2}$
loss function (Section 5.2.3)	ID loss
LR scheduler (Section 5.2.7)	cosine annealing
warm-up epochs (Section 5.2.7)	10

**Table 7 sensors-22-08651-t007:** Performance of the models: from-scratch vs. pre-trained. The weight decay in * was taken from TransReID [47] and kept for the rest of the experiments.

ID	BNNeck	Loss	LR	Weight Decay	* Pre-Trained *	mAP %	R-1 %
1	✗	$L_{ID}$	$1.6\times 10^{-3}$	$5.0\times 10^{-2}$	✗	15.75	23.42
					✓	14.29	35.63
2			$1.0\times 10^{-3}$	$1.0\times 10^{-4}$ *	✗	43.95	77.11
					✓	63.87	91.12
3		$L_{ID},L_{tri}$	$1.0\times 10^{-3}$	$1.0\times 10^{-4}$	✗	54.67	84.44
					✓	73.12	94.39
4		$L_{ID},L_{tri},L_{cen}$	$2.0\times 10^{-3}$	$1.0\times 10^{-4}$	✗	57.39	87.72
					✓	**78.02**	**96.24**
5	✓	$L_{ID},L_{tri},L_{cen}$	$1.5\times 10^{-2}$	$1.0\times 10^{-4}$	✗	59.71	89.39
					✓	77.41	95.88

**Table 8 sensors-22-08651-t008:** Performance of the models using different loss functions and the same learning rates ($1.0\times 10^{-3}$, $2.0\times 10^{-3}$, and $1.5\times 10^{-2}$).

ID	BNNeck	Loss	LR	mAP %	R-1 %
1	✗	$L_{ID}$	$1.0\times 10^{-3}$	63.87	91.12
			$2.0\times 10^{-3}$	64.77	92.07
			$1.5\times 10^{-2}$	68.91	93.68
2		$L_{ID},L_{tri}$	$1.0\times 10^{-3}$	73.12	94.39
			$2.0\times 10^{-3}$	77.04	96.06
			$1.5\times 10^{-2}$	4.51	12.93
3		$L_{ID},L_{tri},L_{cen}$	$1.0\times 10^{-3}$	76.10	95.35
			$2.0\times 10^{-3}$	**78.02**	**96.24**
			$1.5\times 10^{-2}$	0.94	1.54
4	✓	$L_{ID},L_{tri},L_{cen}$	$1.0\times 10^{-3}$	70.73	94.57
			$2.0\times 10^{-3}$	72.89	94.87
			$1.5\times 10^{-2}$	77.41	95.88

**Table 9 sensors-22-08651-t009:** Performances using the LNNeck and various learning rates.

Neck	LR	mAP %	R-1 %
LNNeck	$1.0\times 10^{-4}$	28.6	58.76
	$1.0\times 10^{-3}$	73.85	95.11
	$1.5\times 10^{-2}$	3.73	11.26
	$1.0\times 10^{-1}$	2.01	5.42

**Table 10 sensors-22-08651-t010:** Performance of the models using different learning rates. The loss function is the same for all experiments.

BNNeck	Learning Rate	mAP %	R-1 %
✗	$1.0\times 10^{-3}$	76.10	95.35
	$1.5\times 10^{-3}$	77.38	95.94
	$1.7\times 10^{-3}$	77.72	96.42
	$1.9\times 10^{-3}$	78.00	**96.90**
	$2.0\times 10^{-3}$	**78.02**	96.24
	$2.1\times 10^{-3}$	77.88	96.30
	$2.3\times 10^{-3}$	6.25	21.69
	$3.0\times 10^{-3}$	6.42	20.91
	$6.0\times 10^{-3}$	5.90	19.30
	$8.0\times 10^{-3}$	3.38	9.95
	$1.5\times 10^{-2}$	0.94	1.54
✓	$1.0\times 10^{-3}$	70.73	94.57
	$2.0\times 10^{-3}$	72.89	94.87
	$7.0\times 10^{-3}$	74.94	95.11
	$8.0\times 10^{-3}$	75.00	95.41
	$9.0\times 10^{-3}$	75.35	**96.72**
	$9.5\times 10^{-3}$	75.15	95.76
	$1.0\times 10^{-2}$	75.67	95.64
	$1.25\times 10^{-2}$	76.44	96.42
	$1.45\times 10^{-2}$	76.73	96.00
	$1.5\times 10^{-2}$	**77.41**	95.88
	$1.55\times 10^{-2}$	75.98	95.70
	$1.75\times 10^{-2}$	75.52	95.23
	$2.0\times 10^{-2}$	75.37	95.70

**Table 11 sensors-22-08651-t011:** Performance of the models using the three losses, without the BNNeck and different optimizers and learning rates.

Optimizer	LR	mAP %	R-1 %
**SGD**	** $2.0\times 10^{-3}$ **	**78.02**	**96.24**
AdamW	$2.0\times 10^{-3}$	0.75	1.19
	$2.0\times 10^{-4}$	70.09	93.44
	$7.0\times 10^{-5}$	73.22	94.27
	$2.0\times 10^{-5}$	74.32	94.93
	$2.0\times 10^{-6}$	63.52	87.18
RMSProp	$2.0\times 10^{-3}$	0.73	0.89
	$2.0\times 10^{-4}$	68.74	92.55
	$2.0\times 10^{-5}$	73.67	94.75
	$2.0\times 10^{-6}$	65.41	89.33

**Table 12 sensors-22-08651-t012:** Performance of the D1 and D2 models using different batch sizes.

ID	BNNeck	LR	Variant	Batch Size	mAP %	R-1 %
1	✗	$2.0\times 10^{-3}$	D1	128	77.23	96.72
				256	**78.02**	96.24
2			D2	128	76.60	95.94
				256	76.18	95.41
3	✓	$1.5\times 10^{-2}$	D1	128	73.99	95.58
				256	77.41	95.88
4			D2	128	77.06	96.24
				256	77.16	**97.02**

**Table 13 sensors-22-08651-t013:** Performance of the models using different VOLO variants. ^†^ refers to the models with unstable learning.

Model	Variant	# Params	# Layers	Batch Size	Runtime (h)	mAP %	R-1 %
BN, LR = 0.015	D1	26.6 M	18	256	11.05	77.41	95.88
	D2	58.7 M	24	256	16.68	77.16	97.02
	D3 ^†^	86.3 M	36	128	24.12	75.18	95.88
	D4	193 M	36	128	31.69	78.77	96.66
	D5	296 M	48	128	44.29	**80.30**	97.13
LR = 0.002	D1			256	10.72	78.02	96.24
	D2			128	18.08	76.60	95.94
	D3			128	24.40	76.19	94.93
	D4			128	32.02	78.51	96.78
	D5			128	44.68	79.12	**97.19**

## Data Availability

The dataset VeRi-776 can be requested via https://vehiclereid.github.io/VeRi/ (accessed on 14 May 2022).

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
