# Peer review of "V2ReID: Vision-Outlooker-Based Vehicle Re-Identification"

_sensors, 2022, doi:10.3390/s22228651_

Round 1
Reviewer 1 Report
Authors propose an outlook attention-based vehicle ReID framework using the Vision Outlooker as its backbone, that is able to encode finer-lever features. The paper explains process of implementation of the V2ReID and show that the outlook attention is beneficial to the vehicle recognition task.
Some comments arise after reviewing the paper:
· In the section 'Proposed V2ReID' the sub-sections generally present the state of the art, however the proposed method has not been well developed. The architecture should be discussed in the ‘Proposed V2ReID’ section.
· To show an example of Data-Augmentation (DA) one should use the same image before and after DA to identify the transformations (Figure 10).
· The structure of the paper needs to be reviewed. For example, the 'Experiments and Results' section may contain only two sub-sections 'Implementation Details' and 'Results'.
· The paper requires revision to remove redundancy and unnecessary information.
Author Response
To whom it may concern,
Please see the attachment.
Thank you,
Yan

Reviewer 2 Report
To whom it may concern,
The paper presents the application of vision outlooker architecture to vehicle re-identification problem. The paper is well organized. However, there are few problems still exists in the manuscript.
1. Minor errors:
- Some acronyms are mentioned without referenced, for example: VOLO in line 94. SOTA line 170.
- “small inter-class similarities and large inter-class differences” (line 5, line 31): should small intra-class …
- Some typos, for example: “loose” in line 206 (loose means not firmly or tightly fixed in place)
- Line 142-144: “soft attention can be used to calculate the gradient” – I do understand what the authors are trying to convey. But, the sentence is incorrect. Soft attention is not used to compute the gradient.
- Topic jumps and lack of explanation: The authors have spent a lot of effort to discuss the architecture of the loss functions in section 3.5 of the paper. However, due to the lack of explanation, reasoning and linking sentences, the whole section becomes fragmented. I think the author should give some of their insights after the each sub-sections to improve the quality of the sections
2. Suggestions:
- Line 43-44: “Hence, this model is very limited and cannot be used for real-life applications.” Then, the authors concluded “Due to these limitations, we focus on closed-set ReID in this paper” in line 53. This created a really bad impression about the research topic of the paper. If it cannot be used in the real-life applications, there is no reason to do the research on this topic. The authors need to rewrite this, and give a clear example about the purposes of the research topic.
- Line 165-167: “ However, convolutional filter weights are usually fixed after training. This means that they cannot dynamically adapt to different inputs. Methods using self-attention can alleviate this problem”. Why? Aren't the CNN layers or the FC layers used for attention map prediction also fixed after training? The reasoning here is very unconvincing. The authors need to revise and rewrite this part. To the best of my knowledge, following a neural network layer (CNN or FC) is usually an activation layer such as ReLU. In the case of ReLU, it only preserves the information from the neurons which produce positive values, normally called “activated neurons”. For hard cases, some neurons (which have unimportant information or wrongly activated) can affect the prediction of the neurons in the following layers. The soft-attention can mitigate this by emphasized the important information and minimize the impact of unimportant.
- Line 198: “reshaped into patches” – I don’t think reshaped is the right words in this situation. It’s true that in the pytorch, we can use reshape function to change the shape of a tensor. But, I still think “devided” (as you also used in the manuscript) or “splited” are much more precise in this context.
Author Response

(The authors gave the same response as above.)
